# *Drosophila* as a Model Organism to Understand the Effects during Development of TFIIH-Related Human Diseases

**DOI:** 10.3390/ijms21020630

**Published:** 2020-01-17

**Authors:** Mario Zurita, Juan Manuel Murillo-Maldonado

**Affiliations:** Departamento de Genética del Desarrollo y Fisiología Molecular, Instituto de Biotecnología, Universidad Nacional Autónoma de México, Cuernavaca Morelos 62250, Mexico; jmmurillo.maldonado@gmail.com

**Keywords:** *Drosophila*, TFIIH, human syndromes, Cancer, development

## Abstract

Human mutations in the transcription and nucleotide excision repair (NER) factor TFIIH are linked with three human syndromes: xeroderma pigmentosum (XP), trichothiodystrophy (TTD) and Cockayne syndrome (CS). In particular, different mutations in the XPB, XPD and p8 subunits of TFIIH may cause one or a combination of these syndromes, and some of these mutations are also related to cancer. The participation of TFIIH in NER and transcription makes it difficult to interpret the different manifestations observed in patients, particularly since some of these phenotypes may be related to problems during development. TFIIH is present in all eukaryotic cells, and its functions in transcription and DNA repair are conserved. Therefore, *Drosophila* has been a useful model organism for the interpretation of different phenotypes during development as well as the understanding of the dynamics of this complex. Interestingly, phenotypes similar to those observed in humans caused by mutations in the TFIIH subunits are present in mutant flies, allowing the study of TFIIH in different developmental processes. Furthermore, studies performed in *Drosophila* of mutations in different subunits of TFIIH that have not been linked to any human diseases, probably because they are more deleterious, have revealed its roles in differentiation and cell death. In this review, different achievements made through studies in the fly to understand the functions of TFIIH during development and its relationship with human diseases are analysed and discussed.

## 1. Introduction

The transcription of genes that encode messenger RNAs, (mRNAs), long non-coding RNAs (lncRNAs) and micro-RNA (miRNA) precursors in eukaryotic cells is conducted by the RNA polymerase II (RNPII) enzyme [1]. However, transcription initiation performed in a regulated manner at a precise site in the promoter by RNPII requires assembly of the pre-initiation complex (PIC) [2]. The PIC is composed of a series of factors that are initially recruited by transcription activators in sequence. In general, the PIC is composed of the TFIID complex, which includes TATA-binding-protein (TBP), RNPII, TFIIB, TFIIA, TFIIF, TFIIE, and TFIIH [2,3]. Among the PIC components, the TFIIH complex participates in transcription by RNPII and RNA polymerase I (RNPI), as well as in DNA repair, and one of its subcomplexes participates in cell cycle control (Figure 1). The core and cyclin-dependent-activating-kinase (CAK) subcomplexes [4,5] comprise TFIIH. The core contains the ATPases/helicases XPB and XPD as well as the p62, p52, p44, p34 and p8 subunits and participates itself in the mechanism of nucleotide excision repair (NER). Cdk7, CycH and MAT1 constitute the CAK complex [6]. The core complex participates in the mechanism of nucleotide excision repair, and the ATPase and helicase activities of XPB and XPD are necessary to open double-stranded DNA to facilitate excision of nucleotides from the damaged strand [7]. Furthermore, the CAK complex is required for the phosphorylation of several cyclin-dependent kinases that modulate the cell cycle [4]. In transcription, the CAK complex and core form the 10-subunit TFIIH factor [7]. During transcription activation, the XPB subunit uses its ATPase activity to rotate and translocate DNA into the RNPII, allowing the enzyme to locate the transcription initiation start site to incorporate the first nucleotide to synthetize RNA [8]. Simultaneously, Cdk7 phosphorylates serine 5 of the heptapeptide repeat in the carboxy-terminal domain (CTD) of the RNPII catalytic subunit [9]. This CTD modification is required for RNA processing and modifications and recruitment of the enzyme that introduces the cap to nascent mRNA [10].

Interestingly, drugs that target some of the enzymatic functions of TFIIH have been found or developed. For instance, the drug triptolide, a diterpene tripoxide produced by the *Tripterygium wilfordii* plant used in traditional Chinese medicine, targets the XPB subunit by covalently binding the ATPase domain and therefore inhibits its translocase and helicase activities [11,12]. Furthermore, substances that specifically inhibit the kinase activity of Cdk7 have been developed; one of these drugs is THZ1, which, together with triptolide, has promising prospects for use against cancer [13,14,15,16].

In addition to all the important roles of TFIIH in transcription, DNA repair and cell-cycle control in eukaryotic organisms, TFIIH is also relevant to human health, since mutations in some of its subunits are related to three complex human syndromes. Mutations in the XPB, XPD and p8 subunits are linked to the generation of xeroderma pigmentosum (XP), Cockayne syndrome (CS) and trichothiodystrophy (TTD). Mutations in XPB and XPD may generate XP syndrome; patients with XP syndrome are highly sensitive to UV irradiation and prone to develop skin cancer [17,18,19,20,21]. In addition, XP patients suffer neurodegeneration among other developmental abnormalities. In some cases, XPB and XPD mutations in humans manifest in a combined phenotype reflecting XP and CS; these patients present mental retardation, dwarfism, cachexia and progeria [22]. Mutations in XPB, XPD and p8 may also generate TTD [22] It has been reported that p8 mutations in humans cause a reduction in the level of the TFIIH complex, suggesting that p8 functions to stabilize the complex [22,23,24]. Individuals afflicted with TTD suffer from brittle hair and nails, ichthyosis, mental retardation, dwarfism and osseous deformations [19]. Cells derived from patients with these three syndromes are highly sensitive to ultraviolet (UV) irradiation, indicating a direct effect on the NER mechanism [17,19].

Due to the different functions in which TFIIH is involved, it has been challenging to associate the different manifestations of human syndromes related to TFIIH with particular molecular processes. The molecular nature of the mutations linked to each of these syndromes in the corresponding TFIIH subunit has been identified in many human patients [25,26]. In the case of the helicases XPB and XPD, most of the mutations are located in regions related to their enzymatic activities [25,26]. Based on this information, elegant studies conducted by the Jean-Marc Egly group in Strasbourg using in vitro transcription and NER assays with reconstituted human TFIIH complexes harbouring wild-type XPB and XPD or mutant versions of XPB and XPD found in patients have allowed us to relate syndrome-specific mutations with either transcription or DNA repair. However, these studies were conducted in vitro or in cultured cells, and the effects of these mutations on organism development and physiology can be only extrapolated. Then, mouse models to study the effects of mutations in the p8, XPD and XPB TFIIH subunits were developed and used to understand how these mutations identified in humans affect mouse physiology. Intriguingly, p8 knock-out (KO) in mice was embryonic lethal; therefore, it was difficult to interpret how p8 KO affects development [27]. A mouse model with alterations in XPB that cause a combination of XP and CS in humans showed only partially defective NER and hypersensitivity to UV in the eyes and skin [28]. On the other hand, mouse models of XP and CS in which XPD was mutated were more informative, since in addition to an increased sensitivity to UV irradiation, the mice developed skin cancer, neurodegeneration and cachexia [29]. In addition, a mouse model of TTD in which XPD was mutated presented TTD-like brittle hair and accelerated ageing [30,31]. These mouse models have been useful for gaining an enhanced understanding of the effects of these TFIIH-related syndromes; however, the complexity of mammalian development makes the interpretation of pathological defects difficult. In this situations, an organism such as *Drosophila* has been useful to not only understand how defects in TFIIH that affect transcription and DNA repair impact the development of the organism but also study the effect of mutations in other TFIIH subunits that have not been related to human diseases, probably because they have a more deleterious effect on mammal development. In the following sections, we will analyse and discuss the contribution of *Drosophila* to understanding how alterations in the functions of TFIIH affect tissues at different developmental stages and adult tissues.

## 2. The Phenotypes of TFIIH Mutants in *Drosophila* and Their Relationships with Human Syndromes

The use of *Drosophila* to study homozygous organisms carrying lethal alleles has been possible, allowing the analysis of the defects at different stages of development. This has been the case for the use of different mutant alleles of the TFIIH subunits. The first characterization of a mutant allele of a TFIIH subunit gene was the *haywire* (*hay*) gene, which encodes the *Drosophila* XPB subunit. This allele, named *hay^nc^*, was identified in a genetic screen for genes that interact with testis-specific tubulin and generate male sterility [32]. In addition, several revertant mutants in which the *hay* phenotype in the testis was suppressed were isolated and identified as point mutations of the same *hay* gene [33]. The *hay^nc2^* allele and all the revertants were highly sensitive to UV irradiation, as heterozygous mutants, and *hay*^nc2^ behaved as an antimorphic mutation, while all revertants were hypomorphic and homozygous lethal [33,34]. However, the combination of different *hay* alleles was semilethal, allowing adult organisms to be obtained [34]. These flies presented abdominal defects, abnormal wings and bristle deformations. The abdominal defects were due to a reduction in the thickness of the cuticular layer; in other words, the cuticle was thinner, which was somewhat similar to the ichthyosis phenotype observed in TTD patients. In addition, the defective bristles were thinner and severely deformed, similar to the brittle hair phenotype present in TTD-afflicted individuals. Furthermore, the introduction of mutations in XPB observed in patients into the *hay* gene under a mutant *hay* heterozygous background enhanced these phenotypes [34]. Additionally, a genetic interaction was observed between *hay* mutant alleles and a conditional *cdk7* mutant for these phenotypes, which was the first demonstration of a functional interaction between two TFIIH subunits in a complete animal. Both the cuticle and bristle phenotypes were shown to correlate with a reduction in the transcript levels of genes involved in cuticle formation and, together with the genetic interaction between *hay* and *cdk7* mutants, strongly suggested that these phenotypes are linked to transcriptional defects [34]. Although the skin and hair in humans are not analogous structures to the cuticle and bristles in the fly, the construction of these structures requires high levels of terminal differentiation gene transcription, suggesting that when transcription is reduced but not completely abolished, the most obvious phenotypes are caused by a reduction in the expression in highly transcribed genes. Intriguingly, defects in the development of the nervous system were also observed in *hay* mutants [32,34], demonstrating that these defects are correlated with an increase in apoptosis during fly brain development. Thus, these studies showed that defects similar, but not identical, to those observed in humans can be studied in the fly.

*Drosophila* has also allowed the study of mutations in TFIIH subunits that have not been related to human diseases, probably because their effects are more deleterious and these subunits have a structural and regulatory role in TFIIH. One such example is the p52 subunit. The *marionette* (*mrn*) gene encodes the p52 homologue in *Drosophila*, and initial studies using a combination of alleles showed phenotypes similar to those observed in *hay* mutants, particularly cuticle and bristle defects [35]. In addition, flies in which p52 was mutated were smaller, presenting a minute-like phenotype, that may be linked to a global reduction in transcription, interestingly as previously mentioned, patients afflicted with CS and TTD have a short stature [35]. Additionally, flies in which p52 was mutated developed melanotic tumours correlated with the presence of chromosomal aberrations during development. Importantly, characterization of the nature of different p52 mutant alleles allowed the generation of these mutations in human p52, which were introduced into insect culture cells, co-expressed with all the other components of the TFIIH complex and assayed in in vitro experiments to determine the effect of these mutations on transcription and DNA repair [35]. These experiments confirmed that p52 is important for the incorporation of XPB into the complex and that it modulates XPB-ATPase activity, thereby affecting DNA repair and transcription. Thus, the analysis of p52 mutants in *Drosophila* contributed not only to finding cellular defects generated during development when TFIIH is not functional but also to understanding the mechanistic role of this subunit in the functions of TFIIH.

The implementation of the UAS-GAL4 system in Drosophila allows an easy study of gene functions. Using this system, it was possible to direct specific interfering RNAs against TFIIH subunits in particular cell types. Using this system against the p52 and p34 subunits in the wing imaginal disc, a reduction in the size and number of the cells that generated smaller wings compared with wild-type wings was shown [36]. Intriguingly, a dramatic increase in apoptosis was observed following depletion of the p52 or p34 subunit of TFIIH, and simultaneous depletion of the tumour suppressor p53 enhanced Jun kinase pathway-dependent apoptosis [36]. Furthermore, these phenotypes were phenocopied by the inhibition of XPB ATPase activity with the drug triptolide [36]. This finding is relevant for the interpretation of the link between TFIIH and the p53 and JNK pathways in the generation and treatment of cancer and confirmed that the inhibition of ATPase activity by triptolide causes defects identical to those observed in TFIIH mutants [36].

The p8 TFIIH subunit is enigmatic. When TFIIH was purified from human cells and yeast, all components of the complex were visualized by SDS-polyacrylamide gel electrophoresis, and 9 subunits were identified; however, p8 was not visualized because despite its presence in the gels, as p8 is a 72 amino acid protein, it was not stained. In addition, rare cases of TTD in which neither XPB nor XPD was mutated were identified. A mysterious gene was still missing and named TTDA [37]. It was not until a metaproteomic analysis was conducted in yeast cells that a new protein of approximately 8 kDa was identified as a component of the PIC [38]. Since mutations in the corresponding gene in yeast were highly sensitive to UV irradiation, p8 was believed to be related to the TFIIH complex and at the same time identified as part of the TFIIH complex in humans [23]. Reduced TFIIH levels is characteristic of cells derived from TTDA patients, and the p8 subunit was proposed to stabilize TFIIH [23]. In fact, some reports indicated that in TTDA-derived cells, the levels of ectopic XPB-GFP expression were reduced compared with wild-type cells [27].

In *Drosophila* and in all arthropods, p8 is encoded as a bicistronic transcript with a subunit of the SWIR complex [39]. Thus, the identification of mutants that affect only p8 was complicated. However, an insertion of the P element into half of the p8-coding sequence did not affect the expression of the other proteins, generating a null allele for p8. Thus far, this is the only mutant allele of p8 reported in *Drosophila*. Homozygous flies for this allele are viable; however, males are sterile and present a minute-like phenotype, similar to p52 mutants [39,40]. Interestingly, heteroallelic combinations of p52 alleles are semi-viable, but the males are also sterile. These phenotypes allowed the detailed characterization of defects in spermatogenesis in TFIIH mutants. Intriguingly, p8 and p52 mutations have a moderate effect on gene expression during spermatogenesis and arrested cell differentiation in meiosis, a phenotype that is identical to that in testis-specific mutants of TBP-associated factors (tTAFs). Importantly, in p8 null organisms, the levels of the rest of the TFIIH complex subunits were shown not to be reduced; however, in the p52 mutants, a clear reduction in the XPB and p8 subunits was evident [40]. These results are in contrast with the proposed role of p8 as a stabilizing factor for the TFIIH complex in human cells. However, the p8 mutants derived from patients afflicted with TTDA are not null [27] and seem to act as antimorphs that affect the rest of the TFIIH complex. Instead of p8, the p52 subunit seems to be important to maintain the levels of XPB and p8 [40]. Taking all of the information in this section into account, the analysis of TFIIH mutants in *Drosophila* has allowed a better understanding of the effects of the TFIIH complex during development linked to the different manifestations observed in humans afflicted with XP, CS and TTD.

## 3. Involvement of TFIIH in Cell-Cycle Control, Chromosome Instability and Cancer in *Drosophila*

As mentioned before, the CAK subcomplex of the TFIIH complex independently participates in cell cycle control. For some time, the direct role of Cdk7 as a Cdk-activating kinase (CAK) was elusive. In *Drosophila*, the necessity of Cdk7 for CAK activity was demonstrated in vivo, since temperature-sensitive mutants were defective in activation of the cdc2/Cyc A and cdc2/Cyc B complexes and therefore exhibited defects in cell division [41]. However, it was also shown in *Drosophila* that dominant negative mutants of *cdk7* delayed the onset of transcription in the early embryo but did not affect the fast and synchronized nuclear cycles at the syncytial blastoderm [42].

In addition, the dual role of the CAK subcomplex in cell-cycle control and RNPII transcription has been the subject of investigations on how the CAK subcomplex is regulated. Initially, it was suggested the XPD subunit of the core subcomplex of TFIIH controls the cell cycle function of Cdk7 [43]. This hypothesis was based on the fact that a tetramer composed of CAK and XPD was identified in metazoan cells and that the overexpression of XPD in early *Drosophila* embryos generated a decrease in the T-loop phosphorylation in other Cdk proteins as well as mitotic defects; in contrast, a reduction in XPD levels caused an increase in CAK activity and cell proliferation [43].

Further studies on the possible regulatory role of XPD in CAK activity showed that the complete absence of XPD in early embryos caused defects in the formation of the mitotic spindle as well as changes in the distribution of the CAK subcomplex in different subcellular compartments [44]. Intriguingly, in mammalian cells, XPD may also be part of the MMXD complex, which includes MMS19, Ciao, ANT2 and MIP18, factors that are known to participate in proper chromosome segregation [45]. As described in the same report, abnormal nuclei and mitosis were observed in cells derived from patients with XP and CS with XPD mutations [45]. Furthermore, in *Drosophila* XPD was shown to interact with MMS19, Galla1 and Galla2, which are homologues of MIP18, and the product of *crumbs* [46]. Mutations in these components also generate defects in mitosis in the early embryo, similar to those thought to be caused by deregulation of the CAK subcomplex of the TFIIH complex due to mutations in XPD. However, a recent report showed that defects in mitosis caused by null mutations in MMS19 in *Drosophila* could be rescued by overexpressing the CAK subcomplex or reducing the levels of XPD [47]. Additionally, this study suggested that MMS19 binds XPD and that the CAK complex can phosphorylate Cdk1 [47]. These latter results may explain the interconnection between XPD, the components of the MMXD complex and the CAK subcomplex. However, how XPD knows to interact with MMS19, the CAK subcomplex or the TFIIH complex is not yet understood. In addition, the recent 3D structure of the TFIIH core shows the relevant structural role of XPD in the TFIIH complex [5]. Within the TFIIH complex, XPD makes a strong connection with the p62 and p44 subunits [5]. XPD also directly interacts with XPB and MAT1. Therefore, the absence of XPD may have a deleterious effect on the functions of the TFIIH complex in NER and transcription, thereby complicating the interpretation of defects in mitosis and chromosome segregation and limiting these interpretations only to possible negative modulation by the CAK subcomplex. Indeed, it was recently reported that mutations in p8 and *hay* (XPB) in *Drosophila* cause defects in mitosis and chromosomal segregation in early embryos that are identical to the phenotypes observed in XPD mutants that were attributed to deregulation of the CAK subcomplex [48]. In addition, the knock-down of Cdk7 in the early embryo generated the same phenotypes [48]. Furthermore, the severity of mitotic defect phenotypes in embryos from p8 null mothers was correlated with the direct effect on global gene expression during oogenesis, suggesting that these defects are due to the deficient transcription of maternal genes that participate in fast and synchronized mitosis in the early embryo. Although these defects in mitosis may be caused by the accumulation of a pleiotropic effect on the different functional roles of XPD, the direct role of XPD in regulating the localization and activity of the CAK subcomplex remains controversial.

The fact that mutations in TFIIH subunits generate mitotic defects and chromosome instability links TFIIH with cancer. In addition, patients with XP have an increased predisposition to skin cancer. *Drosophila*, then, is an interesting model in which to study the association between cancer and TFIIH. All the mutants flies in which core subunits of the TFIIH complex were mutated were shown to be sensitive to UV irradiation with different penetrance. Intriguingly, the XPD mutations that generated the highest degree of UV sensitivity in *Drosophila* were not correlated with XPD mutants in humans that generate a higher predisposition to cancer [49]. On the other hand, the mutations associated with a higher risk of cancer in humans in the fly clearly affected the association of XPD with the TFIIH core and CAK subcomplexes and generated a higher frequency of mitotic defects [49]. These results are in agreement with the fact that mutation of the TFIIH complex in general generates catastrophic mitosis.

In addition, in *Drosophila*, it was demonstrated that the TFIIH complex plays a direct role in regulating the fly homologue of human *MYC*. *MYC* in humans is a proto-oncogene that is overexpressed in a large number of different types of tumours, and part of its role in oncogenesis is to over-activate a large number of genes [50]. One of its regulators in human cells is fuse interacting repressor (FIR), a protein involved in mRNA processing and transcription. FIR recognizes a specific element known as the far upstream sequence element (FUSE) to negatively regulate the expression of MYC [51]. Part of the mechanism by which FIR represses the expression of MYC is its interaction with XPB, which tethering FIR to the TFIIH complex and affects formation of the PIC [51]. Half print (Hlp) is the fly homologue of FIR, and in *Drosophila*, Hlp acts as a tumour suppressor; furthermore, mutations in this gene generate cellular overproliferation due to XPB-dependent overexpression of fly *myc* [52,53].

In recent years, the TFIIH complex has become an important target for cancer treatment. This is because it is now evident that cancer cells are addicted to high levels of transcription, and affecting the PIC components may therefore be a suitable way to combat cancer cells [15]. As previously mentioned, the diterpenoid triepoxide triptolide (TLP), derived from *T. wilfordii*, a plant used in traditional Chinese medicine with antiproliferative properties, targets the XPB subunit of the TFIIH complex [12]. TPL inhibits the ATPase activity of XPB, and its derivative minnelide is the subject of a clinical trial [54]. In addition, THZ1 and related compounds that inhibit the kinase activity of CDK7 by binding a protein region outside of its catalytic domain (Cys312) are very effective in killing different types of cancer cells [55,56]. Since TPL inhibits XPB, it has been used to study RNPII transcription initiation in cells and fly embryos [57,58]. However, the effects of neither TPL nor THZ1 have been analysed in cancer models in *Drosophila*. Therefore, the loss of apico-basal polarity by a reduction in the levels of *disc large 1* (*dlg1*), which is involved in basolateral junctions [59], combined with a *ras* mutant allele was used as a model in which severe carcinomas in the wing disc were generated (Figure 2A). Then, we analysed the effect of TPL on these tumours in the wing disc. Intriguingly, the administration of TPL at 5 µM in the food of third instar larvae increased the number of apoptotic cells in the tumour but not in wild-type tissue (Figure 2A–C). In addition, TPL reduced the size of the tumours (Figure 2D). Interestingly, in the tumour cells in the wing disc, the expression of *wingless* (*wg*) was dysregulated, but in the discs from similar larvae fed TPL, the expression pattern of *wg* was partially recovered (Figure 2F,G). These experiments show that the use of the fly as a model organism in which to analyse the effect of drugs that affect the activities of TFIIH in cancer cells is very promising.

## 4. Analysis of TFIIH Complex Dynamics at the Onset of Transcription at the Mid-Blastula Transition

Zygotic genome activation (ZGA), one of the more intriguing stages in animal development, is the moment when gene expression is activated for the first time [60]. During ZGA, thousands of genes are turned on simultaneously [60]. *Drosophila* has been an excellent model in which to study how this process takes place, and there are excellent reviews on this subject [for a recent review: 62]. The early fly embryo is a syncytium that initially suffers 8 fast and synchronized nuclear divisions without transcription conducted by maternal factors deposited in the egg [61]. After nuclear division 8, at the pre-mid blastula transition (pMBT), the first wave of transcription takes place, which activates approximately 100 genes [62], and the rate of nuclear division is increased. At nuclear division 13, the mid-blastula transition (MBT) occurs, and the second wave of transcription takes place, in which more than 3000 genes are activated [62]. Transcription of the first zygotic genes is activated by the transcription factor zelda (zld) with the help of the GAGA factor [63]. To activate transcription, these two factors recruit the PIC at the promoters of genes activated at ZGA, which includes the genes encoding TFIIH complex members. Therefore, this developmental stage in the fly facilitates the study of the recruitment of components of the basal transcription machinery for the first time during development. Studies on the dynamics of the PIC components began with the visualization of RNPII phosphorylated at its CTD by immunostaining, which revealed that it can be detected in somatic nuclei in the syncytial blastoderm only after nuclear division 8 [64] More recently, by using ChIP-nexus experiments, the occupancy of RNPII at the promoters of genes activated at the pMBT and MBT was determined [62]. Another PIC component initially analysed in early embryos was TATA-binding protein (TBP). Interestingly, TBP was found to be imported from the cytoplasm to the nuclei until ZGA [64].

In addition to RNPII and TBP dynamics and the functions of components of the PIC, only studies of TFIIH complex subunits have been performed in the early fly embryo. The first studies on the TFIIH complex suggested that most of the core subcomplex components of the TFIIH complex were cytoplasmic and that these components were localized in the nuclei before the first wave of transcription [65]. However, more refined studies using transgenic organisms expressing several TFIIH subunits fused with different florescent proteins have demonstrated that all the TFIIH subunits, including the CAK components, follow a similar dynamic pattern of expression in the syncytial blastoderm [48]. The TFIIH subunits oscillate between the nuclei and cytoplasm during synchronized nuclear division and are more concentrated in the nuclei. However, during mitosis, TFIIH can still be visualized on chromosomes, suggesting that some TFIIH complex is retained in chromatin due to the fast nature of nuclear division in the early embryo [48]. Intriguingly, in the interphase nucleus and during early mitosis, the TFIIH components can be observed as foci or grains that are similar in appearance to nuclear structures identified as liquid–liquid phase separation condensates Figure 3 [48]. It would be relevant to determine whether functional TFIIH is encased in these condensates and whether some of its subunits facilitate the formation of these membrane-free organelles.

These studies in *Drosophila* are important for understanding the role of TFIIH during development and to relate the TFIIH complex with the functions that are affected in cancer and human syndromes since mammalian models are more complicated and expensive.

## 5. Concluding Remarks and Perspectives

*Drosophila* is a suitable model to analyse the effect of genetic alterations related to human diseases. In the particular case of TFIIH-related syndromes, it has been possible to study defects in the fly produced by mutations in different subunits at a level that cannot be attained in mammal models. In addition, interpretation of the complex genotype–phenotype relationship has been less complicated in *Drosophila*, particularly in phenotypes that are generated during the development of the organism, such as the increase in apoptosis in the nervous system and catastrophic mitosis in the early embryo and larval tissues. In addition, phenotypes in *Drosophila* similar to some of the manifestations of TTD or CS in patients, such as a thin cuticle, brittle bristles and the minute phenotype, were clearly shown to be caused by problems in transcription. Furthermore, in *Drosophila*, it has been possible to study mutations in other TFIIH complex subunits that appear to be highly deleterious in mammals. However, there are still several aspects of this complex that can be studied in *Drosophila* to complement work performed in other models and humans. For instance, it is necessary to analyse mutations in other core TFIIH subunits, such as p34 and p44, and compare the related phenotypes with the reported phenotypes following XPB, XPD, p52 and p8 mutation. Additionally, it would be very informative to determine the genetic interactions of the TFIIH complex with other PIC components and to analyse double mutants in CAK subunits with mutations in the TFIIH core. The TFIIH complex also interacts directly with several transcriptional activators in human cells, the homologues of which in *Drosophila* have been analysed by genetic interaction studies, and their analysis during development will be highly informative for understanding the function of several transcription activators during development. Finally, the performance of chemical genetic experiments by introducing substances that inhibit the enzymatic activities of the TFIIH complex in embryos will allow the role of these specific activities in different developmental processes to be determined.

## Figures and Tables

**Figure 1 ijms-21-00630-f001:**
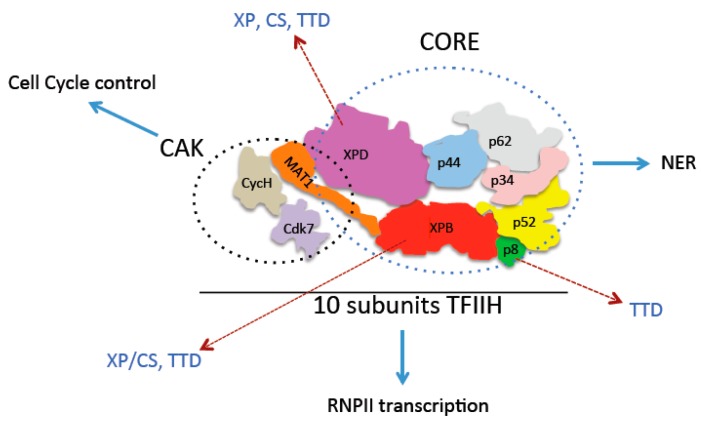
The TFIIH complex, its subunits and links with human diseases. The TFIIH cartoon model is based in the recently published high-resolution structure of the core subcomplex of TFIIH with the MAT1 subunit [5]. Cdk7 and CycH cartoon is fictitious. The name of each subunit is indicated and the core and cyclin-dependent-activating-kinase (CAK) subcomplexes are delimitated by blue and black dashed circles. The functions of TFIIH are also indicated as cell-cycle control for the CAK, and nucleotide excision repair (NER) for the core and transcription by the 10 subunits TFIIH. Also, the subunits affected in the Xeroderma Pigmentosum (XP), Cockeyne Syndrome (CS) and Ticothiodistrohy (TTD), are indicated with dashed arrows.

**Figure 2 ijms-21-00630-f002:**
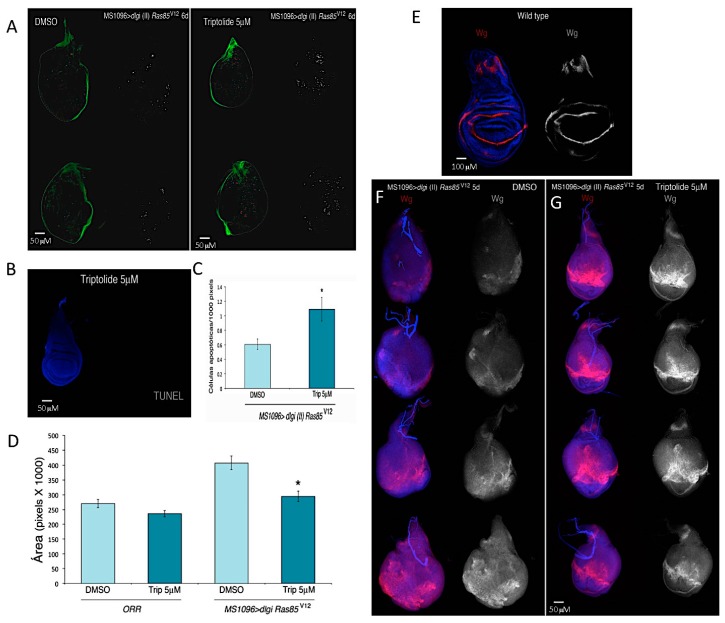
Effect of triptolide (Trip) on third instar wing imaginal disc tumours. (**A**) Shows the generation of apoptosis (TUNEL assay, used to visualize apoptotic cells) by Trip on wing imaginal disc tumours, induced by the knock down (KD) of the *disc large 1* (*dlg*) gene using the MS1096 driver in a *Ras85^v12^5d* mutant allele background (*MS1096>dgl(II) Ras85^v12^5d*). In green the *dlg* expression is visualized and in red the apoptotic cells. Note that the number of apoptotic cells is increased in the tumour tissues treated with Trip. (**B**) TUNEL assay to determine apoptosis in a wild type disc treated with Trip. Note the absence of apoptotic cells. (**C**) Quantification of apoptotic cells from *MS1096>dgl(II) Ras85^v12^5d* treated with DMSO or with Trip. (**D**) Total area of third instar imaginal discs, *wild type* (*ORR*) and *MS1096>dgl(II) Ras85^v12^5d* from third instar larvae feed with DMSO or Trip. Note that the total area in tumour discs is higher than in ORR as well as reduction of the total area in tumour wing discs from larvae feed with Trip. (**E**) Expression pattern (immunostaining) of *wingless* (*wg*) in a wild type third instar larvae wing disc. *wg* is indicated in pink and DNA in blue. (**F**) Expression of *wg* (pink) in *MS1096>dgl(II) Ras85^v12^5d* tumour wing discs, DNA is in blue, from three instar larvae feed with DMSO. Note that the distribution of *wg* is heterogeneous. (**G**) Distribution of *wg* in *MS1096>dgl(II) Ras85^v12^5d tumour wing discs, from three instar larvae feed with Trip.* Note the suppression on *wg* distribution in the discs as well as the reduction of the tumour size.

**Figure 3 ijms-21-00630-f003:**
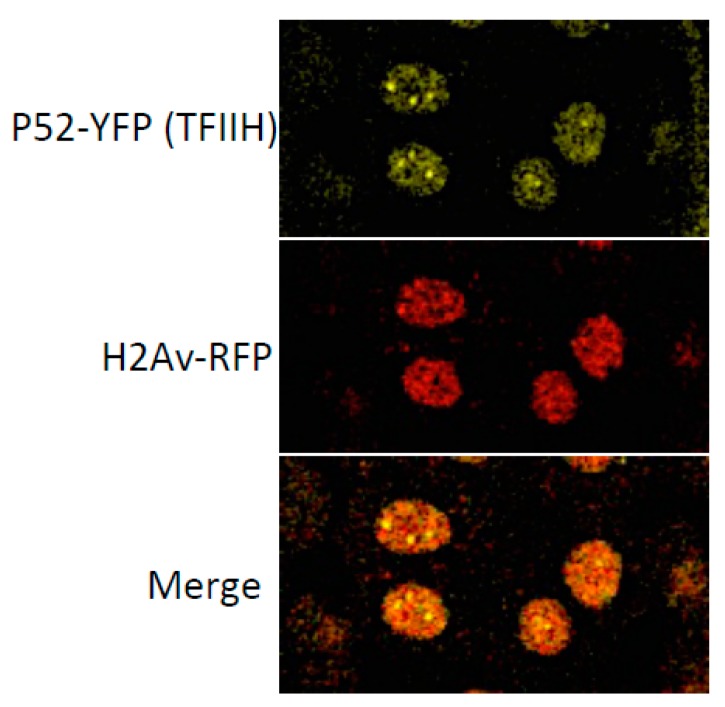
Confocal microscopy images (65X) of syncytial blastoderm nuclei from embryos expressing the p52 subunit of TFIIH fused to the yellow florescent protein (p52-YFP). As control to visualize the chromatin the histone 2Av (H2Av) fused to the red fluorescent protein is show in the same nuclei. Note the presence of p52-YPP foci that resemble liquid-liquid phase separation condensates.

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
