# Peer review of "Drosophila as a Model Organism to Understand the Effects during Development of TFIIH-Related Human Diseases"

_ijms, 2020, doi:10.3390/ijms21020630_

Round 1

Reviewer 1 Report

This manuscript by Mario Zurita and Juan Maneul Murilo-Maldonada summarizes the different achievement through studies in human and Drosophila about the roles of TFIIH and describes the homologous function of TFIIH in mammals and fruit fly. The authors show that the homologous phenotypes between Drosophila, which has been a useful model organism for studying various aspects of biological issues, and humans. The analysis of TFIIH function in development has been limited and so this review progresses our current understanding of this protein. This manuscript is enjoyable to read and conclusions are well supported by their provided evidence and references.

The roles of TFIIH have important roles in transcription, DNA repair and control cell cycle in eukaryotic organisms, TFIIH is also relevant to human health, since mutations in some of its subunits are related to three complex human syndromes: xeroderma pigmentosum (XP), trichothiodystrophy (TTD)and Cockayne syndrome (CS). The complexity of mammalian development makes the interpretation of pathological defects difficult. The analysis of TFIIH function in development has been limited and so this review progresses our current understanding of this protein.

This manuscript is enjoyable to read and conclusions are well supported by their provided evidence and references. The referee recommends publication of this manuscript in IJMS.

Author Response

We want to thank the reviewer for the time and help in the review of this manuscript.

Reviewer 2 Report

This is a manuscript reviewing the contribution of Drosophila melanogaster as model system to study TFIIH and the impact of its mutant alleles on transcription and DNA repair. The manuscript is well-structured and referenced, and offers a comprehensive view of the current knowledge on this component of the pre-initiation complex.

I would like to suggest few changes (reported below) that will possibly improve the quality of this manuscript:

line 39 NER should be defined upon its first use in the text

lines 41-42 proposed change: "excision of the damaged strand" to "excision of nucleotides from the damaged strand"

line 101 "In the Drosophila field, it is believed that any gene mutation generates a phenotype...." I don't think anybody using Drosophila as a genetic model could believe this. Please consider revise this statement.

line 103 "...the use of Drosophila to study homozygous organisms carrying lethal alleles allows analysis of the penetrance..." This sentence is incorrect in this form. Please revise.

lines 138-139 Does the minute phenotype result only from mutations in the p52 gene? Why mutations in genes encoding other subunits of the PIC complex do not cause the same phenotype? The minute phenotype in Drosophila is usually linked to translation defects. Could be this clarified?

line 150 "An advantage to the use of Drosophila to study gene functions is the UAS-GAL4 sys" this sentence could be misinterpreted. I would personally change it as follow "The implementation of the UAS-GAL4 system in Drosophila allows an easy study of gene functions".

Author Response

Here we enclose the answers, point by point of the reviewer concerns. We want to thank the reviewer comments that substantially improve this review.

1.- line 39 NER should be defined upon its first use in the text.

This point is clarified in the new version fo the article as follows:

“..participates itself in the mechanism of Nucleotide Excision Repair (NER)”.

2.- lines 41-42 proposed change: "excision of the damaged strand" to "excision of nucleotides from the damaged strand"

We thank the reviewer for this observation, and it has been changed as follow:

“. DNA to facilitate excision of nucleotides from the damaged strand”

3.- line 101 "In the Drosophila field, it is believed that any gene mutation generates a phenotype...." I don't think anybody using Drosophila as a genetic model could believe this. Please consider revise this statement.

We agree with the reviewer. This sentence has been deleted and now the paragraph initiates as follows:

The use of Drosophila to study homozygous organisms carrying lethal alleles has been possible, allowing the analysis of the defects at different stages of development

4.- line 103 "...the use of Drosophila to study homozygous organisms carrying lethal alleles allows analysis of the penetrance..." This sentence is incorrect in this form. Please revise.

As it is indicated in the previous point this has been changed as follows:

The use of Drosophila to study homozygous organisms carrying lethal alleles has been possible allowing the analysis of the defects at different stages of development. This has been the case for the use of different mutant alleles of the TFIIH subunits..

5.- lines 138-139 Does the minute phenotype result only from mutations in the p52 gene? Why mutations in genes other subunits of the PIC complex do not cause the same phenotype? The minute phenotype in Drosophila is usually linked to translation defects. Could be this clarified?

This is a very good observation by the reviewer. In fact, also the p8 mutant also presents this minute like phenotype. In the new version of this manuscript it has been changed as follows:

In line 137: In addition, flies in which p52 was mutated were smaller, presenting a minute like phenotype, that may be linked to a global reduction in transcription, interestingly as previously mentioned, patients afflicted with CS and TTD have a short stature [35].

And in line 177: Homozygous flies for this allele are viable; however, males are sterile and present a minute like phenotype, similar to p52 mutants [40, 41].

6.- line 150 "An advantage to the use of Drosophila to study gene functions is the UAS-GAL4 sys" this sentence could be misinterpreted. I would personally change it as follow "The implementation of the UAS-GAL4 system in Drosophila allows an easy study of gene functions".

We thank the reviewer for this observation and the sentence has been changed as it is suggested.

We want to thank the reviewer for the time and help in the review of this manuscript.